# Possible Significance of Neutrophil–Hemoglobin Ratio in Differentiating Progressive Supranuclear Palsy from Depression: A Pilot Study

**DOI:** 10.3390/diseases13040119

**Published:** 2025-04-18

**Authors:** Michał Markiewicz, Natalia Madetko-Alster, Dagmara Otto-Ślusarczyk, Karolina Duszyńska-Wąs, Bartosz Migda, Patryk Chunowski, Marta Struga, Piotr Alster

**Affiliations:** 1Department of Neurology, Medical University of Warsaw, Kondratowicza 8, 03-242 Warsaw, Poland; natalia.madetko@wum.edu.pl (N.M.-A.); karolina.duszynska@gmail.com (K.D.-W.); patryk.chunowski@wum.edu.pl (P.C.); 2Department of Biochemistry, Medical University of Warsaw, Banacha 1, 02-097 Warsaw, Poland; dagmara.otto@wum.edu.pl (D.O.-Ś.); marta.struga@wum.edu.pl (M.S.); 3Diagnostic Ultrasound Lab, Department of Pediatric Radiology, Medical University of Warsaw, Kondratowicza 8, 03-242 Warsaw, Poland; bartosz.migda@wum.edu.pl

**Keywords:** depression, inflammation, N/HGBR, NLR, PLR, NHR, atypical Parkinsonism

## Abstract

Background: Research has associated chronic inflammation with the evolution of neurological and psychiatric disorders. Neurodegenerative diseases, including Parkinson’s Disease (PD), Alzheimer’s Disease (AD), and less common ones such as Progressive Supranuclear Palsy (PSP), are commonly linked to depression. However, the pathomechanisms and the role of neuroinflammation in these disorders remain unclear; therefore, interest is increasing in easily accessible inflammatory morphological assessments of blood samples, such as the neutrophil-to-lymphocyte ratio (NLR), the neutrophil-to-monocyte ratio (NMR), and the neutrophil-to-hemoglobin ratio (N/HGBR). Methods: The authors analyzed 15 age-matched controls and 21 patients with PSP; the PSP group was additionally divided into 11 patients without depression (PSP) and 10 with depression (Beck Depression Inventory [BDI] ≥ 14) (PSP-D). Results: In the PSP-D group, the level of N/HGBR was significantly lower than in the controls (*p* = 0.01), but there were no significant differences in any other neutrophil-derived parameters or comparisons of morphological blood assessment. Patients with PSP-D exhibited a marginally significant decrease in neutrophil levels compared to the controls. Conclusions: This is the first study highlighting the possible significance of peripheral inflammatory factors in patients with PSP affected by depression. It highlights possible tendencies in the area of non-specific inflammatory markers and suggests their relation to affective disorders in PSP.

## 1. Introduction

The pathophysiology of Parkinsonian syndromes remains not fully recognized; the clinical picture can exhibit motor and non-motor symptoms (NMSs) [1]. NMSs lead to decreased quality of daily life and disability (disproportionately to motor dysfunction). In patients with Parkinson’s Disease (PD), depression (as the main NMS) is diagnosed in 30–40%, whereas patients with Progressive Supranuclear Palsy (PSP) present a higher frequency of depressive disorders [2,3].

Factors associated with the pathogenesis of these diseases include neuroinflammation, oxidative stress, and mitophagy disruption. In the context of neuroinflammation, the specific impact on the course of Parkinsonism has not been uncovered. Moreover, whether neuroinflammation is a cause or consequence is debated. A meta-analysis of peripheral inflammatory markers revealed elevated interleukin 1 beta (IL-1β), interleukin 2 (IL-2), interleukin 6 (IL-6), tumor necrosis factor-alpha (TNF-alpha), interleukin-10 (IL-10), and C-reactive protein (CRP) levels in patients with PD and atypical Parkinsonisms [4]. Recently, there has been growing interest in unspecific ratios of selected blood cells and high-density lipoprotein, in the context of the peripheral pathological inflammatory response. Markers for chronic subclinical inflammation, such as the neutrophil-to-lymphocyte ratio (NLR), the platelet-to-lymphocyte ratio (PLR), the monocyte-to-lymphocyte ratio (MLR), and the neutrophil-to-high-density lipoprotein ratio (NHR), have been evaluated in various neurodegenerative disorders. In addition to these approaches, many studies on major depressive disorder (MDD) are now focusing on the role of central and peripheral inflammation as diagnostic and prognostic factors.

Major depressive disorders are among the common mood conditions with the highest prevalence in PSP. Specified diagnostic criteria, according to the Diagnostic and Statistical Manual of Mental Disorders (DSM V) by the American Psychiatric Association (2013), are implemented for precise diagnosis, which are also necessary for treatment evaluation [5]. For the assessment of depression among patients with Parkinsonian syndromes, the Beck Depression Inventory II (BDI-II) is a tool of great reliability for correlating higher BDI-II scores with lower quality of life (QoL), irrespective of motor and cognitive symptoms [6].

## 2. Materials and Methods

### 2.1. Group Description

This study analyzed the differences in biochemical parameters between patients with PSP without a diagnosis of MDD (*n* = 11) and those with MDD comorbidity (PSP-D, *n* = 10), and compared them to healthy controls (*n* = 15), who were similar in age and other demographic parameters. Regarding the presence of depression, two subgroups of patients with PSP were categorized based on BDI scores: a PSP (BDI < 14) group and a PSP-D (BDI ≥ 14) group [6,7]. All patients were examined by a neurologist and a neuropsychologist experienced in the assessment of movement disorders at the Department of Neurology, Medical University of Warsaw, between 2020 and 2023. The diagnosis of PSP was based on recent criteria [8]. The patients included in the PSP-D group received a BDI score equal to or higher than 14 points. BDI is a standard method of assessment, enabling the evaluation of the core clinical characteristics of depression, which encompass its negative symptoms; this 21-item self-assessment instrument is one of the most popular measures worldwide [9]. All patients underwent a comprehensive blood analysis at the Laboratory Diagnostics Department of Mazovian Brodno Hospital (Sysmex xs-1000i Hematology Analyzer, by Sysmex Corporation, Kobe, Japan), providing morphological and biochemical evaluations (e.g., neutrophil and lymphocyte count, platelet count); all ratios were calculated using aforementioned patterns based on the blood testing obtained from a single sample.

Patients diagnosed with PSP (without comorbidity of MDD) had a mean age of 68.4 years (SD = 6.3), with ages ranging from 59 to 77 years. Patients in the PSP-D group had a mean age of 72.3 years (SD = 6.2), which was very similar to that of the control group, whose mean age was 71.3 years (SD = 5.6). Both PSP subgroups exhibited male predominance, whereas the control group exhibited a distribution with female predominance, consisting of 10 females and 5 males (Table 1).

Among the exclusion criteria used in this study were neoplasms, chronic inflammatory diseases, autoimmune diseases, infectious diseases, vascular abnormalities, and psychiatric disorders other than depression. The treatment of all patients was analyzed for possible influence on mood disorders, and individuals impacted by drug treatments were excluded. Most of the patients with PSP were not receiving any dopaminergic agents at the time of the study, as these drugs were withdrawn due to their primary or secondary inefficiency. Only treatment for irrelevant comorbidities was deemed eligible for all patients with PSP and healthy controls.

### 2.2. Statistical Analysis

Statistical analyses were conducted using GraphPad software, ver. 9.3 (GraphPad Software, Boston, MA, USA). Variables following a normal distribution were expressed as mean and standard deviation (SD). Comparisons between two groups were performed using the Mann–Whitney U test, while the Kruskal–Wallis test was utilized for comparisons involving three or more groups. If the Kruskal–Wallis test indicated statistical significance, Dunn’s post hoc test was conducted. A two-tailed *p*-value of less than 0.05 was considered statistically significant.

## 3. Results

### 3.1. Neutrophil-to-Hemoglobin Ratio (N/HGBR)

The mean N/HGBR for the PSP cohort was 0.4 (SD = 0.2; range: 0.2–0.6) and was significantly higher than for the PSP-D group (mean score: 0.3; SD = 0.1; range: 0.2–0.4; *p* = 0.01) (Table 1). For parameters with group-to-group differences showing statistical significance, the neutrophil count was also determined (*p* = 0.04) (Table 1).

### 3.2. Neutrophil–Lymphocyte Ratio (NLR)

The mean NLR for the PSP cohort was 2.6 (SD = 1.0; range: 1.6–3.6); compared to the PSP-D group (2.7 ± 1.0), the difference was not statistically significant (Table 1).

### 3.3. Neutrophil-to-Monocyte Ratio (NMR)

The mean NMR for the PSP cohort was 8.6 (SD = 2.7); compared to the PSP-D group (7.9 ± 2.3), the difference was not statistically significant (Table 1, Figure 1).

### 3.4. Correlations Between N/HGBR and Other Inflammatory Markers and Clinical Measures

Pearson’s correlation analyses were performed to explore the relationships between the N/HGBR and various clinical and inflammatory variables, including HGB, LYM, MONO, NEU, NLR, NMR, and BDI. The results of these analyses are summarized in a correlation matrix, with further details presented in a heatmap visualization (Figure 2).

A strong positive correlation was found between the N/HGBR and neutrophil count (NEU) (r = 0.95, *p* < 0.0001), which remained statistically significant after Bonferroni correction (adjusted *p* = 0.0007). This indicates that higher N/HGBR values are strongly associated with elevated neutrophil levels.

A moderate positive correlation was observed between the N/HGBR and the neutrophil-to-lymphocyte ratio (NLR) (r = 0.67, *p* = 0.0247), although this correlation lost statistical significance after Bonferroni correction (adjusted *p* = 0.1729). Similarly, a negative correlation was found between the N/HGBR and hemoglobin levels (HGB) (r = −0.44, *p* = 0.1173), which was also non-significant after correction (adjusted *p* = 0.8211).

The correlation between the N/HGBR and lymphocyte count (LYM) was minimal (r = −0.01, *p* = 0.4897), and this result remained non-significant after correction. A moderate positive correlation between the N/HGBR and monocyte count (MONO) (r = 0.55, *p* = 0.0609) was also observed, but it became non-significant after correction (adjusted *p* = 0.4263). Likewise, the N/HGBR showed a weak positive correlation with NMR (r = 0.45, *p* = 0.1141), which was not statistically significant after the correction (adjusted *p* = 0.7987).

Finally, no significant correlation was found between the N/HGBR and BDI (r = −0.10, *p* = 0.3954), and this result remained non-significant even after correction (adjusted *p* = 2.7678).

## 4. Discussion

The outcomes of this study stress the possible significance of peripheral inflammation in the context of clinical severity. This study reveals that the level of neutrophils in patients affected by PSP with depression is lower than in patients with PSP without depression and in healthy controls. This shows that the combined effect of PSP and depression may be affected by an additional factor inhibiting the stimulation of neutrophils; however, the description of this process is not specified.

Depression is associated with increased levels of neutrophils, as stressed in multiple studies regarding non-specific inflammatory factors such as the neutrophil-to-lymphocyte ratio. The increased levels of this parameter were found to be correlated with the severity of depression and cardiovascular risk factors [10]. In this context, an intuitive outcome of this study suggests that the neutrophils and derived factors, like the neutrophil-to-hemoglobin ratio in the group of PSP with depression, should be increased, which is contrary to the actual outcomes. The results of this study may be partly explained by the impact of two clinical aspects: PSP as the primary disease and depression as the co-existing factor. The pathophysiology of PSP is linked to neuroinflammation; however, a more thorough analysis of the recently discovered mechanisms underlying this disease indicates the possible significance of certain cytokines regarded as anti-inflammatory factors. Evaluations of the M2 phenotype of microglia have indicated its possible relevance in tauopathies at their early stages. M2 is related to the production of IL-4, IL-10, IL-13, and TGF-β, which are classified as anti-inflammatory cytokines [11]. In this context, the PSP pathomechanism tends to be a complex process affected by proinflammatory factors, such as IL-1 and IL-6, and the abovementioned anti-inflammatory cytokines [12].

Previous assessments of inflammation in PSP showed a positive correlation between PLR and IL-6 in serum [12]. This may suggest that the microglial activation impacting the levels of IL-6 may play a role in the evolution of certain groups of symptoms in atypical Parkinsonism. Peripheral inflammatory parameters have yet to be used to examine this group. Initially, studies concerning this assessment were based on the analysis of NLR [13]. They indicated the possibly elevated levels of NLR among patients with PSP in comparison with patients with Parkinson’s Disease. The examination of PSP and CBS did not show any significant differences in NLR. Patients with PSP-RS had higher levels of NLR than the control group [14]. A meta-analysis on the significance of NLR showed higher levels of neutrophils among patients with PSP and not significantly deviated lymphocytes compared to controls [15]. The activated leukocyte metabolism in PSP and NRF2/HO-1 pathway activation were linked with accumulation in peripheral blood mononuclear cells [16]. The features were found to be negatively correlated with the PSP-rating scale. Examinations concerning the association between peripheral inflammatory parameters and neuroimaging revealed a negative correlation between NHR and perfusion in the insula and thalamus only among patients with CBS [17], which has not been confirmed in PSP. The results obtained from these studies suggest that the inflammatory impact may differ depending on the region of interest in neuroimaging.

Despite multi-dimensional studies, the exact causes of depressive disorders are still unknown, and it is believed that the co-occurrence of various endogenic (genetic predisposition, physical brain structure changes, psychological factors, comorbidity, gender) and environmental factors contributes to MDD [18,19,20]. Those may be additionally influenced by lifestyle factors (like a highly processed diet leading to gut dysbiosis, poor physical activity, or psychoactive substances use) [21,22]. The progression of the mentioned neurodegenerative disorders worsens with MDD and other mood changes, significantly affecting patients’ QoL.

Currently, although not fully discovered, the multi-systemic and entangled pathophysiology pathway of affective disorders includes neurotransmission alterations in monoamines and glutaminate, neurotrophin imbalance, or hypothalamic–pituitary–adrenal axis dysregulation, followed by immune system dysfunctions and local inflammation [23]. Neuroinflammation in mood disorders is currently being intensively explored to uncover promising discoveries that can improve their treatment [24]. Interestingly, a bidirectional relationship has been reported: depression increases inflammatory responses, which in turn increase depression [25]. For neuroinflammation evaluation, numerous peripheral and central biomarkers have been assessed for their feasibility, e.g., C-reactive protein (CRP), several interleukins, cortisol, and tumor necrosis factor-alpha (TNF-α), among others. A complex pathway of inflammatory responses is initiated with neutrophil stimulation, impacting the production of numerous non-specific mediators, leading to phagocytic and apoptotic effects. Moreover, lymphocytes are responsible for immune system regulation and suppression. Thus, neutrophil numbers refer to an unspecific inflammatory process, whereas lymphocyte counts are caused by physiological stress [26]. These ratios are highly accessible due to availability from routine complete blood count, at low costs. Also, for mood disorders and other psychiatric comorbidities, their use is well established [27]. Most studies, including meta-analyses, have indicated that NLR levels are associated with the severity of depression [28,29]. However, in some studies, no significant difference in NLR between patients with depression and controls was found, but the researchers assumed that this was due to the relatively small group samples [30]. In another study, the correlation with MDD severity was not significant in a female subgroup, which was explained by changes in estrogen levels. However, other scientists doubt this explanation, stressing the need for further studies in this field [31,32]. Data from a meta-analysis on inflammation and depression reveal a noticeable heterogeneity across studies [33]. Despite suggestions for future evaluation, NLR is recognized as suitable for neuroinflammation assessment, but as an unspecified marker, it is also useful as a diagnostic biomarker and predictor of several mood disorders presence, including MDD and psychotic depression [34,35].

Although antidepressants influence neuroinflammation, some data suggest that they do not affect NLR [36,37]. Because it is recognized to be less affected by confounding conditions, NLR may be more informative, compared to other leukocyte parameters or non-leukocyte-based markers. Other markers have not been as well explored as NLR yet. In some studies, NLR and PLR were found to be correlated with depression severity, but other researchers did not find such a correlation, although some suggested a possible link between the clinical profile of MDD with PLR, but not with NLR [31,38]. All individuals diagnosed with MDD showed significantly increased PLR in comparison with healthy controls [39]. A meta-analysis of different studies revealed a significant association between PLR, NLR, and MLR values [31]. The importance of immune dysregulation in mood disorders (including MDD) and its possible treatment target was revealed in randomized controlled studies. The addition of anti-inflammatory drugs (i.e., statins, celecoxib, and omega-3 fatty acids) to a selective serotonin reuptake inhibitor (SSRI) resulted in greater efficiency in MDD treatment, compared with SSRI monotherapy [40,41].

This study has several limitations. As all patients included in the study were alive, no neuropathological examination was performed, and a definitive diagnosis was not accessible. The diagnosis of PSP was based on probable and possible diagnoses according to the most contemporary criteria [8]. The examination was based on non-specific assessment methods of peripheral inflammation; however, the authors intended to evaluate factors with high accessibility, low cost, and minimal invasiveness that were feasible to assess in clinical practice. This study was based on a single examination of patients, which excluded the analysis of time-related tendencies.

The outcomes of this research suggest that depression may be a factor additionally enhancing one of the contrary mechanisms. This study’s methodological limitations and characteristics as a pilot study limit the interpretation and applicability of the results. Understanding the co-existing factors impacting the pathophysiology of PSP seems crucial in achieving future optimal therapeutic strategies; in this context, pilot studies highlighting such tendencies are valuable.

## 5. Conclusions

To the best of our knowledge, this is the first study indicating the possible significance of the inflammatory process in the pathogenesis of both PSP and depression. The pathomechanism of this Parkinsonian syndrome remains unexplained; thus, further research is recommended, preferably including larger groups of participants. As neuroinflammation seems to be one of the common underlying causes for both mood disorders and some neurodegenerative syndromes, there is a need for further exploration. Advances in preclinical studies may lead to the identification of binding points for new drugs with potentially disease-modifying effects.

## Figures and Tables

**Figure 1 diseases-13-00119-f001:**
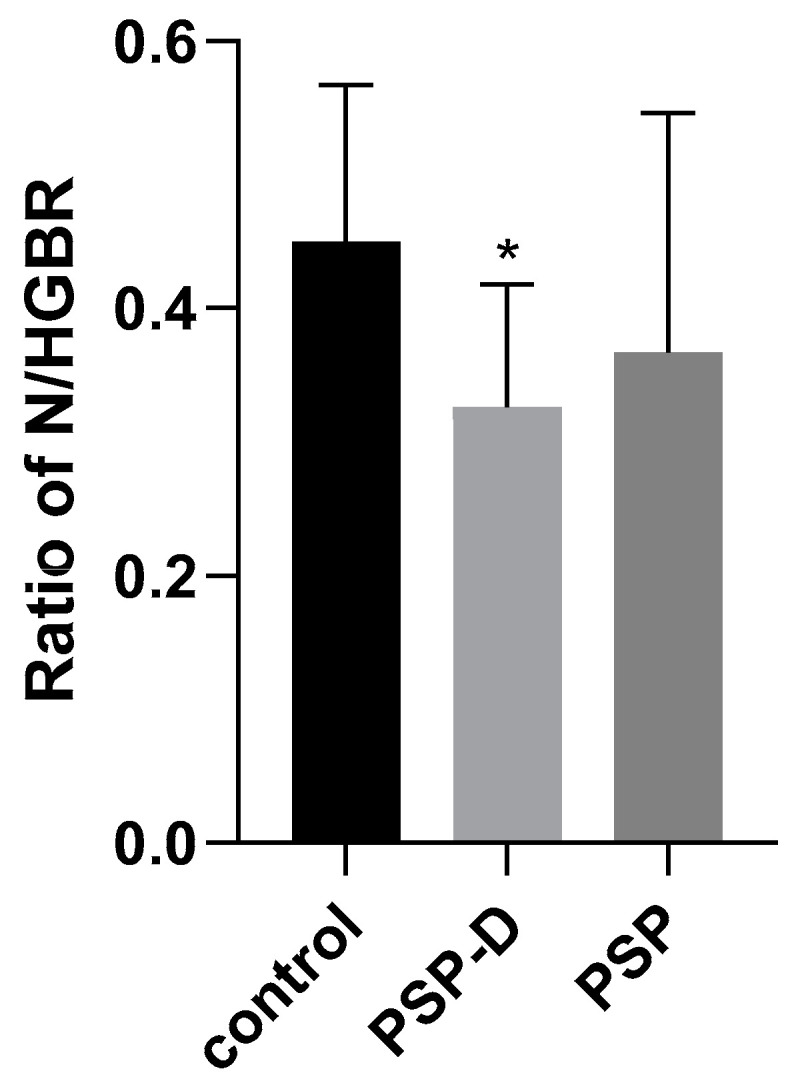
Level of N/HGBR in serum. Comparisons were carried out between the Progressive Supranuclear Palsy (PSP) and PSP with depression (PSP-D) groups and controls. Statistical significance was calculated using analysis of variance followed by Dunne’s post hoc test, * *p* < 0.01. Legend: N/HGBR—neutrophil-to-hemoglobin ratio; PSP—Progressive Supranuclear Palsy; PSP-D—PSP with depression.

**Figure 2 diseases-13-00119-f002:**
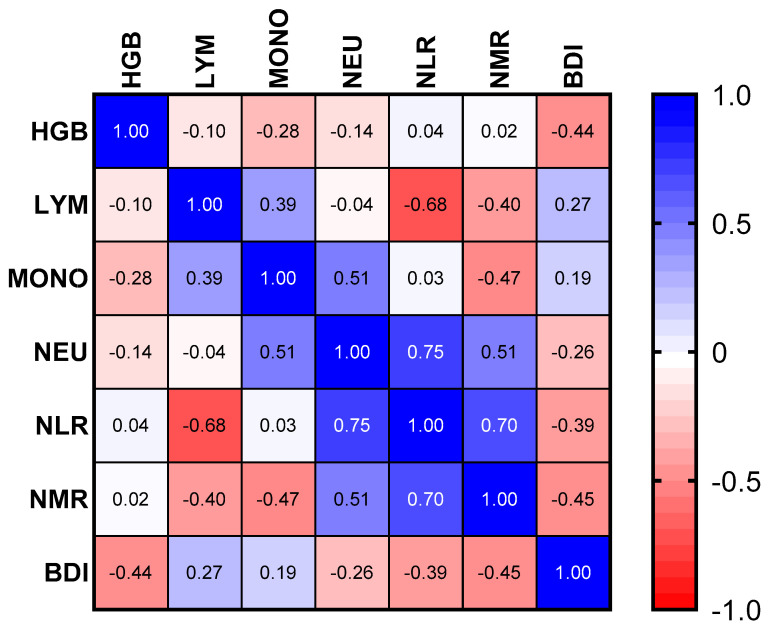
Heatmap showing Pearson’s correlation coefficients between the N/HGBR and various clinical and inflammatory markers. The color intensity represents the strength of the correlation, with blue indicating negative correlations, red indicating positive correlations, and white indicating no correlation.

**Table 1 diseases-13-00119-t001:** Descriptive statistics of analyzed parameters.

Variables	Control (*n* = 15)	PSP (*n* = 11)	PSP-D (*n* = 10)	*p*-Value
	Mean ± SD	Mean ± SD	Mean ± SD	
Sex (female/male)	10/5	2/9	4/6	-
Age	71.3 ± 5.6	68.4 ± 6.3	72.3 ± 6.2	-
NEU [×10^9^/L]	6.0 ± 1.9	5.6 ± 1.9	4.4 ± 1.1	0.04 ^a^
HGB	13.4 ± 1.2	14.2 ± 1.5	13.8 ± 1.0	-
LYM [×10^9^/L]	2.0 ± 0.9	2.1 ± 0.7	1.7 ± 0.5	-
MONO [×10^9^/L]	0.6 ± 0.2	0.6 ± 0.4	0.6 ± 0.1	-
N/HGBR	0.44 ± 0.1	0.4 ± 0.2	0.3 ± 0.1	0.01 ^a^
NLR	4.4 ± 4.1	2.6 ± 1.0	2.7 ± 1.0	-
NMR	12.8 ± 9.4	8.6 ± 2.7	7.9 ± 2.3	-
BDI	5.3 ± 2.4	9.4 ± 2.9	20.7 ± 5.1	-

Statistical significance was calculated using the Mann–Whitney U-test. Significant *p*-value (*α* = 0.05). ^a^, significant for PSP × control; Legend: *p*—*p*-values for Student’s *t* test; BDI—Beck Depression Inventory (PSP group—BDI scores below 20; PSP-D group—BDI scores of 20 and above); N/HGBR—neutrophil-to-hemoglobin ratio; NLR—neutrophil-to-lymphocyte ratio; NMR—neutrophil-to-monocytes ratio; NEU—neutrophil; HGB—hemoglobin; LYM—lymphocytes; MONO—monocytes.

## Data Availability

Data sharing is not applicable.

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
