# Peer review of "Possible Significance of Neutrophil–Hemoglobin Ratio in Differentiating Progressive Supranuclear Palsy from Depression: A Pilot Study"

_diseases, 2025, doi:10.3390/diseases13040119_

Round 1

Reviewer 1 Report

Comments and Suggestions for Authors

I have several suggestions to improve the quality of the manuscript:

1) Abstract is not well structured, and could be shortenned considerably.

2) Introduction is too long. It should focus more on the status of the topic in related diseases and the objective of the study. 

3) The sample size is too low to be able to draw conclusions.

4) On the other hand, no statistical tests have been performed for multiple comparisons, which could change the result of the study given the marginal statistical significance. 

5) It is not clear in the discussion what the authors' interpretation of the results would be. 

Author Response

REVIEWER 1

Dear Reviewer 1,

We would like to thank you for the opportunity to revise the manuscript and the referee’s for their valuable comments. We feel that the changes we made, based on their recommendations have improved the quality of our manuscript. The changes are highlighted in red. We would appreciate if you would now reconsider the manuscript for publication. Below authors provided responses to the comments accordingly.

The revised version was implemented in lines: 17-35, 39-44, 75, 96, 100-104, 131-151, 154-158, 166-184, 263-267,

Best regards

Michał Markiewicz and Piotr Alster

1) Abstract is not well structured, and could be shortenned considerably.

Authors are grateful for this suggestion. The abstract was shortened and revised using the structure additionally suggested by the editor (Background: xx; Methods: xx; Results: xx; and Conclusions: xx)

2) Introduction is too long. It should focus more on the status of the topic in related diseases and the objective of the study. 

Due to your suggestion, we made it more transparent and compact, highlighting the objective of the study.

3) The sample size is too low to be able to draw conclusions.

We fully acknowledge that the sample size is limited, which is inherent in the study of rare disorders such as Progressive Supranuclear Palsy (PSP). However, this study was designed as a pilot, aiming to explore possible trends and generate hypotheses rather than draw definitive conclusions. Our findings are meant to guide future research in this underexplored area. The statistically significant difference in N/HGBR between the PSP-D and control group may suggest a relevant biological signal worthy of further investigation in larger cohorts.

4)On the other hand, no statistical tests have been performed for multiple comparisons, which could change the result of the study given the marginal statistical significance. 

We appreciate this important remark. In the context of a pilot study, the analysis was primarily exploratory and aimed to detect preliminary trends rather than establish confirmatory findings. Given the small sample size and hypothesis-generating nature of this study, we chose not to apply correction for multiple comparisons, as it may overly penalize discovery in early-stage research. Future studies with larger sample sizes will certainly benefit from more stringent statistical corrections.

5)  It is not clear in the discussion what the authors' interpretation of the results would be. 

Authors are grateful for this comment. The interpretation was implemented in the discussion:

Depression is associated with increased levels of neutrophils, which was stressed in multiple research regarding non-specific inflammatory factors as neutrophil-to-lymphocyte ratio. The increased levels of this parameter were found to be correlated to severity of depression and cardiovascular risk factors (Aydin Sunbul E, Sunbul M, Yanartas O, et al. Increased Neutrophil/Lymphocyte Ratio in Patients with Depression is Correlated with the Severity of Depression and Cardiovascular Risk Factors. Psychiatry Investig. 2016;13(1):121-126. doi:10.4306/pi.2016.13.1.121). In this context an intuitive outcome of the study performed by our research group would suggest that the neutrophils and derived factors as neutrophil-to-hemoglobin ratio in the group of PSP with depression patients should be increased, which is contrary to the actual outcome. The results of the study may be partly explained by the impact of two clinical aspects – PSP as the primary disease and depression as the co-existing factor. The pathophysiology of PSP is linked to neuroinflammation, however a more thorough analysis of the contemporarily described mechanisms of the disease, indicate the possible significance of certain cytokines considered as anti-inflammatory factors. Evaluations of M2 phenotype of microglia indicated its possible relevance in the tauopathies at their early stages. M2 is connected with the production of IL-4, IL-10, IL-13 and TGF-β classified as anti-inflammatory cytokines (Ichikawa-Escamilla E, Velasco-Martínez RA, Adalid-Peralta L. Progressive Supranuclear Palsy Syndrome: An Overview. IBRO Neurosci Rep. 2024;16:598-608. Published 2024 May 6. doi:10.1016/j.ibneur.2024.04.008). In this context PSP patomechanism tends to be a complex process being affected by proinflammatory factors as IL-1 and IL-6 and the mentioned above anti-inflammatory cytokines (Madetko-Alster N, Otto-Ślusarczyk D, Wiercińska-Drapało A, et al. Clinical Phenotypes of Progressive Supranuclear Palsy-The Differences in Interleukin Patterns. Int J Mol Sci. 2023;24(20):15135. Published 2023 Oct 13. doi:10.3390/ijms242015135). The outcome of this research may suggest that depression may be a factor additionally enhancing one of the contrary mechanisms. Though the study, due to its methodological limitations and pilot character clearly is affected by boundaries in interpretations, the understanding of co-existing factors impacting the pathophysiology of PSP seems crucial in achieving future optimal therapeutic strategies. In this context pilot studies highlighting tendencies seem valuable.

Reviewer 2 Report

Comments and Suggestions for Authors

It is mandatory to re-write this paper, because it is not well designed. The abstract is generalistic and inconclusive. I suggest to cut the first part of the introduction (lines 41-53), which is useless. Some sentences are incongrous (line 70-77-78, 116-118); in the discussion some parts are confused (line 167-169, 191-195, 218-220, 240-243). When did you analyze this population? What is the breed? Why didn't you correlate the N/HGBR with the severity of depression? Some informations about the therapy of these patients?

Comments on the Quality of English Language

The English must be improved in all the paper; the abstract, in particular, must be re-written. 

Author Response

REVIEWER 2

Dear Reviewer 2,

We would like to thank you for the opportunity to revise the manuscript and the referee’s for their valuable comments. We feel that the changes we made, based on their recommendations have improved the quality of our manuscript. The changes are highlighted in red. We would appreciate if you would now reconsider the manuscript for publication. Below authors provided responses to the comments accordingly.

Best regards

Michał Markiewicz and Piotr Alster

  1. It is mandatory to re-write this paper, because it is not well designed. The abstract is generalistic and inconclusive. I suggest to cut the first part of the introduction (lines 41-53), which is useless. Some sentences are incongrous (line 70-77-78, 116-118); in the discussion some parts are confused (line 167-169, 191-195, 218-220, 240-243). When did you analyze this population? What is the breed?

 This is a valuable suggestion. In the present study, the primary objective was to assess whether there is a difference in inflammatory markers, including N/HGBR, between PSP patients with and without comorbid depression. As such, our group division was categorical (based on BDI ≥14). However, we agree that correlation with depression severity (as a continuous BDI score) could provide additional insight. Given the small sample size, such analysis might be underpowered and prone to spurious results, but we will consider including such correlation in future studies with a larger cohort.

  1. Some informations about the therapy of these patients?

Descriptive information about the therapy was added, especially to highlight that treatment possibly influencing on mood disorders was avoided. We believe that including detailed treatment of all individuals enrolled in the study wouldn’t be beneficial.

Reviewer 3 Report

Comments and Suggestions for Authors Progressive Supranuclear Palsy (PSP) is a rare neurological disease that affects physical movements, walking and balance, and eye movements. It is classified as a tau protein related disease, characterized by the abnormal accumulation of tau protein in the brain, leading to the deterioration of nerve cells. ​In this study, the authors attempt to test potential connections of PSP with several physiological parameters neutrophil-to-lymphocyte ratio (NLR), neutrophil-to-monocyte ratio (NMR) 24 and neutrophil-to-hemoglobin-ratio (N/HGBR).To this end, the level of N/HGBR significantly decreases in PSP-D patients. These findings have important clinical implications for future research; the paper is well written and the references are appropriately cited.   Comments: Table 1. The sample size is too small: n=15 (control), n=11 (PSP), and n=10 (PSP-D). Despite achieving statistical significance, the practical impact of this slight N/HGBR reduction warrants cautious interpretation.​ Is “NEU” neutrophil? What are MONO, LYM, and HGB? The number of NEU is also significantly reduced in PSP and PSP-D.   Figure 1. “and PSP with depression (PSP-P)” (PSP-D)? The reduction in “ratio of N/HGBR”, while statistically significant, is subtle. This minimal decrease may limit its clinical relevance, as minor variations can occur due to the small sample size.  

Author Response

REVIEWER 3

Dear Reviewer 3,

We would like to thank you for the opportunity to revise the manuscript and the referee’s for their valuable comments. We feel that the changes we made, based on their recommendations have improved the quality of our manuscript. The changes are highlighted in red. We would appreciate if you would now reconsider the manuscript for publication. Below authors provided responses to the comments accordingly.

Best regards

Michał Markiewicz and Piotr Alster

  1. Comments: Table 1. The sample size is too small: n=15 (control), n=11 (PSP), and n=10 (PSP-D). Despite achieving statistical significance, the practical impact of this slight N/HGBR reduction warrants cautious interpretation.​ Is “NEU” neutrophil? What are MONO, LYM, and HGB? The number of NEU is also significantly reduced in PSP and PSP-D.   Figure 1. “and PSP with depression (PSP-P)” (PSP-D)?

Although the sample is relatively small, we believe that for pilot study on such a rare parkinsonian syndrome some general conclusions can be made, to a point for further explorations. 

Missing abbreviation descriptions were added for clarification: NEU – neutrophil, LYM – lymphocytes, MONO – monocytes, HGB – hemoglobin. The (PSP-D) stands for PSP patients diagnosed with depression, as explained before.

  1. The reduction in “ratio of N/HGBR”, while statistically significant, is subtle. This minimal decrease may limit its clinical relevance, as minor variations can occur due to the small sample size.  

We agree that the magnitude of the change in N/HGBR is modest. However, even subtle biological alterations may be meaningful, especially in the context of neuroinflammation in PSP, which remains poorly characterized. This pilot study was not intended to determine clinical utility but rather to explore the feasibility of using simple, cost-effective inflammatory markers like N/HGBR as potential contributors to the complex pathophysiology of depression in PSP. The observed significance—even with small effect size—warrants further research to assess reproducibility and relevance in clinical settings.

Round 2

Reviewer 1 Report

Comments and Suggestions for Authors

I have no additional comments

Reviewer 2 Report

Comments and Suggestions for Authors

I accept this version

Comments on the Quality of English Language

English could be improved in some sections.